# STOP-Bang questionnaire should be used in all adults with Down Syndrome to screen for moderate to severe obstructive sleep apnea

Anderson Albuquerque de Carvalho[1,2☯¤*], Fabio Ferreira Amorim[3‡], Levy Aniceto Santana[3‡], Karlo Jozefo Quadros de Almeida[4‡], Alfredo Nicodemos Cruz Santana[2,3☯], Francisco de Assis Rocha Neves[1☯]

1 Department of Postgraduate Health Sciences, Faculdade de Ciências da Saúde, Universidade de Brasília, Brasília, Distrito Federal, Brazil, 2 Respiratory Department, Multidisciplinary Sleep Unit, Hospital Regional da Asa Norte, Brasília, Distrito Federal, Brazil, 3 Department of Postgraduate Health Sciences, Faculdade de Medicina e Enfermagem, Escola Superior de Ciências da Saúde, Brasília, Distrito Federal, Brazil, 4 Regional Board of Secondary Care Department, Down Syndrome Reference Center, Hospital Regional da Asa Norte, Brasília, Distrito Federal, Brazil

☯ These authors contributed equally to this work.
¤ Current address: Respiratory Department, Multidisciplinary Sleep Unit, Hospital Regional da Asa Norte, Brasília, Distrito Federal, Brazil
‡ These authors also contributed equally to this work.
* carvalhofisio2003@gmail.com

**Data Availability Statement:** All files used to write the manuscript are available in a public database

## Abstract

### Study objectives

To determine the prevalence of obstructive sleep apnea (OSA) in adults with Down syndrome (DS), to investigate factors related to OSA severity and to identify which sleep questionnaire is the most appropriate for the screening of OSA in this population.

### Methods

Cross-sectional study that consecutively included 60 adults with DS. All patients underwent type III polysomnography and clinical and laboratory data were collected; sleep assessment questionnaires were applied. Multiple linear regression models evaluated the associations between OSA severity (measured by the respiratory event index—REI) and clinical and laboratory data and sleep questionnaires (Epworth Sleepiness Scale, Pittsburgh Sleep Quality Index, BERLIN and STOP-Bang questionnaires).

### Results

Results show that 60 (100%) adults with DS had OSA, with moderate-severe OSA identified in 49 (81.6%). At the multivariate linear regression, REI significantly correlated with hematocrit levels, BMI and STOP-Bang questionnaire (SBQ) results (P <0.001). The positive STOP-Bang ≥3 points) showed 100% of sensitivity (95%CI: 92.75–100%), 45.45% of specificity (95%CI: 16.75–76.62), positive predictive value of 89.09% (95%CI: 82.64–93.34%), negative predictive value of 100%, accuracy of 90% (95%CI: 79.49–96.24%) and OR of 24.29.

repository: https://doi.org/10.6084/m9.figshare.
9788903.v1.

**Funding:** This study was supported by grants from the Fundação de Ensino e Pesquisa em Ciências da Saúde (FEPECS), process number 064.000.560/ 2015, of Public Notice number 40 of 10/29/2015, published in DODF n. 213, of 11/06/2015, regarding the Homologation of the Final Selection Result of Research Projects. The funders had no role in study design, data collection and analysis, decision to publish, or preparation of the manuscript. Fundação de Ensino e Pesquisa em Ciências da Saúde (FEPECS): URL: http://www. fepecs.edu.br/ Homologation of the Final Selection Result of Research Projects: URL: http://www. dodf.df.gov.br/index/visualizar-arquivo/?pasta= 2015/11_Novembro/DODF%20213%2006-11- 2015&arquivo=DODF%20213%2006-11-2015% 20SECAO3.pdf.

**Competing interests:** The authors have declared that no competing interests exist.

## Conclusions

Adults with DS have a very high prevalence of OSA. Hematocrit levels, BMI and SBQ showed a strong correlation with OSA severity. The SBQ performed well in identifying moderate to severe OSA in this population. Considered together, these results point to the need to perform OSA screening in all adults with DS, and STOP-Bang may play a role in this screening.

## Introduction

Down syndrome (DS) is the most prevalent chromosomal abnormality, with approximately 5.4 million affected individuals worldwide [1], with 206,000 of these patients in the United States [2]. In the last decades, the life expectancy of individuals with DS has considerably increased and currently exceeds 60 years in developed countries. [1] A common comorbidity in people with DS is obstructive sleep apnea (OSA), whose prevalence ranges from 78 to 100% in adults [3–8] and 69% to 76% in children [9]. Moreover, when compared to the general population, the OSA is more often classified as severe and is associated with more significant hypoxemia in individuals with DS. [3] In turn, OSA-related hypoxemia has been associated with decreased verbal IQ, executive function, visual-perceptual skills and increase in mood disorders in patients with DS. [10]

The higher prevalence and severity of OSA in patients with DS is related to the phenotypic characteristics of DS itself. These changes include upper airway hypotonia, midface hypoplasia, mandibular hypoplasia, glossoptosis, lingual tonsil hypertrophy, pharyngomalacia, laryngomalacia, and tonsil / adenoid hypertrophy. Additionally, factors such as obesity and hypothyroidism, which are also frequently observed in individuals with DS, favor OSA onset. [3,10,11]

Despite the high risk for OSA in individuals with DS, recommendations for OSA screening and polysomnography in this population are not well established. The American Academy of Pediatrics recommends that polysomnography be performed in all children with DS up to the age of 4, or earlier, if OSA symptoms are present. [12] As for adult patients with DS, the recommendation is also to perform OSA screening, but it is not clearly defined how the screening should be performed (e.g., using a questionnaire or through polysomnography), nor the periodicity of this screening (e.g., annual or biannual assessment). Another important issue is that, for the adult population with DS, there are no validated questionnaires for OSA screening and there is no specific recommendation on which questionnaire should be applied to this special group of patients.

Considering that in the overall population, adequate OSA treatment helps to reduce the incidence of comorbidities and improve neurocognitive functions [13], the identification of OSA in patients with DS is essential to ensure a better neurological prognosis.

Therefore, the aim of the present study was to evaluate OSA prevalence and factors associated with OSA severity in adult patients with DS. Moreover, we investigated which questionnaire is the most appropriate for OSA screening in this population.

## Methods and materials

This study was carried out from October 2017 to October 2018 and included 66 adults with DS attending the Down Syndrome Reference Center (CRISDOWN) of Hospital Regional da Asa

Norte (HRAN) linked to Faculdade de Medicina da Escola Superior de Ciências da Saúde (ESCS), Brasília, Federal District, Brazil.

Subjects were sequentially recruited and submitted to type III polysomnography, anthropometric data collection, laboratory tests and sleep assessment questionnaires.

The inclusion criteria comprised individuals with DS treated at our service, of both genders, good overall health status, aged 18 years and older, capable to understand and accept the study and its procedures. The following were excluded: (1) individuals under 18 years of age; (2) patients undergoing treatment for sleep disorders; (3) patients with a history of conditions that could affect brain structure or function (such as cerebrovascular accident or head trauma); (4) those who refused to participate in the study.

A researcher explained the study in details to all participants with DS and their proxies. Those who agreed to participate in the study signed the free and informed written consent form. The study was approved by the Research Ethics Committee of Fundação de Ensino e Pesquisa em Ciências da Saúde (FEPECS) according to Opinion number 1,656,332/2016 and followed the principles established by the Declaration of Helsinki.

## Data collection and research protocol

After the study participants were defined, a structured interview was carried out to collect the following data: age, gender, BMI, neck circumference (NC), Mallampati class and comorbidities. Venous blood samples were obtained for analysis after a fasting period of 12 to 15 hours. Laboratory tests included complete blood count, thyroid stimulating hormone (TSH), fasting glucose, HOMA-IR, alanine aminotransferase (ALT), uric acid, urea, creatinine and sodium levels.

The following questionnaires were also applied for sleep assessment: Epworth Sleepiness Scale (ESS) [14,15], Pittsburgh Sleep Quality Index (PSQI) [16], BERLIN Questionnaire (BQ) [17,18], and the STOP-Bang Questionnaire (SBQ) [19,20].

In the sleep assessment questionnaires, due to the absence of cutoff points specifically defined for the adult population with DS, the ones defined for the overall population were used, as follows: (1) probable presence of excessive daytime sleepiness in the ESS (total score> 10) [14]; (2) poor sleep quality in the PSQI (total score >5) [16]; (3) risk for OSA in the BQ (two or more positive categories) [17]; (4) risk for OSA in the SBQ (3 or more affirmative answers) [19]. These questionnaires were answered by the patients' proxies, in agreement with other studies in individuals with DS. [4,5,21]

Moreover, all research subjects underwent type III polysomnography according to international guidelines. [22] This type III polysomnography has been previously used in OSA studies in individuals with DS [23] and in other populations [24,25].

In the present study, the equipment used to perform the type III PSG assessments was the ApneaLink Air (ResMed Germany Inc.), which has been previously used in other important studies. [24,25] This equipment allowed monitoring with the use of nasal pressure cannula (for airflow and snore detection), chest piezoelectric strap (for respiratory effort detection) and pulse oximetry (to monitor peripheral arterial oxygen saturation—$SpO_2$ and heart rate).

The total recording time was used as the denominator to calculate the respiratory event index (REI). [22,26] The PSG was assembled by a specialized technician from the Sleep Laboratory of our service. The patient was then referred to a temperature-controlled and quiet bedroom, to undergo the all-night test, without the technician's supervision. All patients were accompanied by a proxy during the examination, who slept in a separate bed, but in the same room. Recording of sleep examination records started at 8pm and stopped at 7am in the following morning.

The respiratory events were defined as follows: (1) hypopnea, when there was a ≥30% reduction in airflow for at least 10 seconds, observed through the nasal cannula, associated with a decrease in $SpO_2$ of at least 3%; (2) obstructive apnea, due to the absence or reduction ≥ 90% of airflow for at least 10 seconds in the presence of respiratory effort; (3) mixed apnea, due to the absence or reduction ≥ 90% of airflow, without the presence of respiratory effort only at the beginning of the event; (4) central apnea, due to absence or ≥ 90% reduction in airflow for at least 10 seconds associated with absence of respiratory effort throughout the event. [27]

The OSA was then classified as mild, moderate or severe. It was classified as mild OSA when the REI was between 5 and 14.9 events / hour, moderate when the REI was between 15 and 29.9 events / hour and severe when the REI was 30 or more events / hour (detected by type III polysomnography). [27]

The polysomnography exams were manually reviewed by a single physician according to AASM standardized criteria. [27] The examiner was blinded to any patient clinical data. The PSG examinations were considered valid when there was ≥ 4 hours of adequate recording without significant loss of sensor signal (nasal pressure cannula, pulse oximetry and chest strap). [28–30]

## Statistical analysis

Numerical data are presented as mean and standard deviation. Categorical data are presented as absolute numbers and percentage. The statistical analysis was initially performed with simple (univariate) linear regression analysis to identify factors correlated with REI. Subsequently, variables with p-values <0.15 were considered for the multiple linear regression analysis, using variable selection according to the backward technique. A p-value <0.05 was used as the criterion for the retention of variables in the final multiple linear regression model. [31,32] Logarithmic transformation was used to obtain the dependent variable distribution normalization. Assumptions of normality, homocedascity (constant variance) and residual independence were assessed. The presence of multicollinearity was assessed by estimating the variance inflation factor (VIF), with VIF >5 being used as an indication of multicollinearity in the multiple linear regression analysis. [32]

The adjusted coefficient of determination $R^2$ was used as a measure of fit quality. Normality was assessed by visual inspection of histograms and by applying the Shapiro-Wilk normality test. [33,34]

The sleep questionnaires that remained in the final multiple linear regression analysis model were used as categorical variable (positive or negative questionnaire) together with moderate to severe OSA (present or absent) in 2x2 contingency tables. Consequently, odds ratio (OR), sensitivity, specificity, positive predictive value (PPV), negative predictive value (NPV) and accuracy were calculated.

The R software (R Foundation, Vienna, Austria) was used for statistical data analysis. [33,34] All hypothesis tests were bilateral. Values of p <0.05 were considered significant.

## Results

During the study period, 66 adults with DS were included in the research. Of this total, 60 completed and 06 patients (9%) dropped out of the study out of their own will. Ten percent (6 subjects) had to undergo the PSG again due to technical problems, such as loss of signal from one of the sensors.

Table 1 shows the participants' demographic characteristics (N = 60). A total of 33 men (55%) and 27 women (45%) participated in the study and the mean age of the sample was

**Table 1. Clinical and laboratory characteristics of adult participants with Down syndrome.**

| Variable | Overall |
|---|---|
| Age, years | 27.7± 9.1 |
| Gender, Male sex | 33 (55%) |
| BMI, kg/m$^2$ | 27.9 ± 5.9 |
| Overweight | 14 (23%) |
| Obesity | 27 (45%) |
| Neck circumference, cm | 40 ± 3.7 |
| Mallampati class, 3–4 | 52 (86%) |
| Hypothyroidism | 40 (66%) |
| Congenital cardiopathies | 30 (50%) |
| Uric acid | 6.2 ±1.5 |
| Creatinine | 0.96 ± 0.18 |
| Glucose | 63.9 ± 34.4 |
| Hematocrit | 47.7 ± 5.4 |
| HOMA-IR | 6.3 ± 6.2 |
| Sodium | 140.1 ± 2.9 |
| ALT | 23.1 ± 10.1 |
| TSH | 2.6 ± 1.8 |
| Urea | 29.6 ± 11.2 |

Values are shown as mean ± SD, N (%). SD = standard deviation, BMI = body mass index, ALT = alanine aminotransferase, TSH = thyroid stimulating hormone. N = 60.

27.7 ± 9.1 years. Obesity and overweight were observed in 27 (45%) and 14 (23%) subjects, respectively. Oropharyngeal examination showed that 52 (86%) participants were Mallampati class III or IV. Clinically, 30 individuals (50%) were diagnosed with congenital heart disease, and 7 had undergone previous surgical correction.

Regarding the questionnaires used for sleep assessment (Table 2) it was observed that: (1) ESS suggested sleepiness (ESS >10) in 32 participants (53%); (2) PSQI identified sleep

**Table 2. Sleep-related characteristics of adult participants with Down syndrome.**

| Variable | Overall |
|---|---|
| REI | 30.4 ± 19 |
| OSA severity | |
| Mild | 11 (18.3%) |
| Moderate | 26 (43.3%) |
| Severe | 23 (38.3%) |
| (1) ESS score | 10.8 ± 3.9 |
| Total score > 10 | 32 (53%) |
| (2) PSQI score | 6.4 ± 2.2 |
| Total score > 5 | 37 (61%) |
| (3) BQ score, risk for OSA | 49 (81%) |
| (4) SBQ score, risk for OSA (≥3) | 55 (91%) |

Values are shown as mean ± SD, N (%). SD = standard derivation, REI = respiratory event index, BMI = body mass index., BQ = Berlin Questionnaire, ESS = Epworth Sleepiness Scale, PSQI = Pittsburgh Sleep Quality Index, SBQ = Stop-Bang Questionnaire. N = 60.

**Table 3. Results of simple (univariate) linear regression analysis for the dependent variable respiratory event index.**

| Covariate | Level | Regression coefficient (Beta) | 95%CI Lower | 95%CI Upper | P-value |
|---|---|---|---|---|---|
| Uric acid | | 0.230 | 0.149 | 0.312 | < .001* |
| Creatinine | | 0.637 | -0.235 | 1.509 | 0.149 |
| Glucose | | 0.024 | 0.011 | 0.036 | < .001* |
| Hematocrit | | 0.082 | 0.062 | 0.102 | < .001* |
| HOMA-IR | | 0.054 | 0.032 | 0.075 | < .001* |
| Age | | 0.014 | -0.003 | 0.031 | 0.102 |
| BMI | | 0.034 | 0.009 | 0.059 | **0.01*** |
| Sodium | | 0.038 | -0.015 | 0.090 | 0.157 |
| ALT | | 0.023 | 0.008 | 0.037 | **0.003*** |
| TSH | | -0.047 | -0.134 | 0.041 | 0.288 |
| Urea | | 0.012 | -0.002 | 0.026 | 0.083 |
| Congenital cardiopathies | Yes | 0.037 | -0.278 | 0.352 | 0.814 |
| | No | - | - | - | - |
| BQ score | Positive | 0.795 | 0.446 | 1.145 | < .001* |
| | Negative | - | - | - | - |
| ESS score | ESS > 10 | 0.081 | 0.046 | 0.116 | < .001* |
| | - | - | - | - | |
| PSQI score | PSQI > 5 | 0.072 | 0.005 | 0.139 | **0.037*** |
| | - | - | - | - | |
| SBQ score | ≥3 | 0.958 | 0.447 | 1.469 | < .001* |
| | 0–2 | - | - | - | - |
| Neck circumference | | 0.051 | 0.010 | 0.092 | **0.016*** |
| Mallampati class | 3/4 | 0.837 | 0.430 | 1.245 | < .001* |
| | 1/2 | - | - | - | - |
| Hypothyroidism | Yes | 0.058 | -0.276 | 0.392 | 0.729 |
| | No | - | - | - | - |
| Number of drugs used | ≥ 2 | 0.312 | -0.051 | 0.675 | 0.091 |
| | 0/1 | - | - | - | - |
| Gender | Female | -0.227 | -0.538 | 0.084 | 0.149 |
| | Male | - | - | - | - |

BMI = body mass index, ALT = alanine aminotransferase, TSH = thyroid stimulating hormone, BQ = Berlin Questionnaire, ESS = Epworth Sleepiness Scale,

PSQI = Pittsburgh Sleep Quality Index, SBQ = Stop-Bang Questionnaire.

*Statistical significance = $p < 0.05$. N = 60 in each of the simple linear regression analyses.

disorders (PSQI >5) in 37 participants (61%); (3) the BQ and (4) the SBQ indicated high risk of OSA in 49 participants (81%) and 55 participants (91%), respectively.

When performing the type III PSG, it was observed that the 60 (100%) study individuals had OSA (REI ≥ 5 events / h), with moderate to severe OSA in 49 (81.6%). Central sleep apnea syndrome (> 5 events / h) was not observed in any of the subjects.

At the simple linear regression analysis (Table 3), several variables showed a significant linear association with the dependent variable (REI), such as uric acid, glucose, hematocrit, HOMA-IR, BMI, ALT, neck circumference, Mallampati class, BQ, ESS, PSQI and SBQ. However, the multiple linear regression analysis (final model), as shown in Table 4, identified only hematocrit, BMI and SBQ as independent predictors of REI. On average, for every 1-unit increase in the hematocrit or BMI variables, and for SBQ to go from negative to positive, the REI response variable increased by 0.073, 0.028, and 0.616 respectively. None of the other

**Table 4. Results of the multivariate linear regression analysis for the dependent variable respiratory event index.**

| Covariate | Level | REI | | | |
| | | Regression coefficient (Beta) | 95%CI Lower | 95%CI Upper | P-value |
|---|---|---|---|---|---|
| Hematocrit | | 0.073 | 0.056 | 0.090 | $< .001^*$ |
| BMI | | 0.028 | 0.013 | 0.043 | $< .001^*$ |
| SBQ | $> = 3$ | 0.616 | 0.289 | 0.943 | $< .001^*$ |
| | 0–2 | - | - | - | - |

REI = respiratory event index, BMI = body mass index, SBQ = Stop-Bang Questionnaire. Adj R-Sq = 0.69.

$^*$ Statistical significance. N = 60.

sleep questionnaire scores (BQ, ESS, and PSQI) demonstrated a significant positive linear association with REI.

As the SBQ remained in the final multiple linear regression model, the contingency table analysis (2x2) with moderate-severe OSA was performed. Positive SBQ (3 or more affirmative answers) showed in the screening of moderate-severe OSA, 100% of sensitivity (95%CI: 92.75–100%), 45.45% of specificity (95%CI: 16.75–76.62%), 89.09% of PPV (95%CI: 82.64–93.34%), 100% of NPV, 90% of accuracy (95%CI: 79.49–96.24%) and OR of 24.29.

## Discussion

The present study contributes with important findings. First, when compared to previously published studies [3–8], it included the largest number of adults with DS. Second, it evaluated for the first time the role of the SBQ in moderate-severe OSA screening in this population. Moreover, this was the first study to evaluate the prevalence of OSA in DS adults in Brazil.

Thus, we demonstrated that the prevalence of OSA in individuals with DS in our sample was 100%. Moreover, 81.6% of the subjects had moderate to severe OSA. Interestingly, participants with DS were included in the present study regardless of whether or not they had significant complaints of sleep-disordered breathing. This fact supports the idea of performing active OSA screening in all adult individuals with DS during routine consultations [3,5]. Compared to other OSA studies in adult individuals with DS, the prevalence of OSA was also high and ranged from 78 to 100%. It is noteworthy that the number of individuals included in such studies was usually small, the smallest with 3 patients and the largest with 47 individuals [3–8]. The difference in OSA prevalence between studies possibly derives from heterogeneities between them, such as different types of polysomnography for OSA diagnosis, as well as sample characterization and size.

Our results also demonstrated that BMI is positively related with OSA severity indicated by the REI. This finding corroborates the high prevalence of the observed moderate-severe OSA, based on the fact that 68% of our adult patients with DS were obese or overweight. This correlation between BMI and REI has been well documented in the overall population [35] and in adults with DS [3,5,6].

Interestingly, the hematocrit level is also related with OSA severity (REI). This relation has been previously identified in individuals in the overall population with OSA [36,37]; however, it had not yet been demonstrated in the population of DS adults. This relation possibly reflects the presence and greater severity of nocturnal hypoxemia that stimulates increased erythropoietin synthesis, with consequent elevated erythrocyte production and increased hematocrit levels [36,37].

Besides, our study also showed the relation between higher SBQ score and higher REI (OSA severity) in DS adults. This relation has been widely shown in the general population

[38–40]. As an example, a recent meta-analysis study demonstrated that the higher the SBQ score, the greater the likelihood of moderate-severe OSA [38].

Regarding the use of sleep questionnaires for OSA screening, our study was the first to identify a questionnaire capable of playing a significant role in moderate-severe OSA screening in DS adults. The sensitivity and accuracy of the SBQ for detecting moderate-severe OSA was 100% and 90%, respectively.

This result indicates that the SBQ should be applied to all adult individuals with DS, thus helping to prioritize individuals who need to undergo the polysomnography. This recommendation becomes important as we observe that the international guidelines indicate OSA screening in adults with DS, but do not clearly indicate how this screening should be performed [41–44].

Finally, regarding the results of the BQ, ESS and PSQI questionnaires not showing a role in relation to OSA in our study, they are in accordance with the results of the study by Giménez et al [5].

Our study shows some limitations. First, the individuals were submitted to a type III PSG and not to a full PSG test. Consequently, we did not have information related to sleep architecture, and OSA severity (REI) may have been underestimated. However, the type III PSG has been used in DS with good results. [23,45–47] Moreover, the number of technical problems that required the type III PSG to be repeated was small (10% in our study), which is in accordance to previous studies in DS or not (3–18%). [22,23,47–49]

Another limitation of the present study is the absence of DS adults without OSA. Since the present study was performed in a single center for DS patients from a reference hospital and did not include patients treated in a community care institution, our results may represent a specific population of DS adults, with more comorbidities. Moreover, considering that the prevalence of DS patients without OSA should be quite low, it will be necessary to increase the sample size, and, at some point, a small number of adults without OSA would be identified. Interestingly, so far, the largest study that evaluated OSA in adults DS patients included 47 individuals [5] reported a prevalence of 78% of OSA. Therefore, the high prevalence of OSA observed in our DS patients is similar to what was described in other studies, which varies between 78–100% [3–8]. It is conceivable that the absence of adults with DS without OSA is possibly related to the high prevalence of overweight and obesity observed in our patients, which was 68%. Nevertheless, a similar prevalence of overweight and obesity was described by other studies [5,50]. The prevalence of OSA observed in the present study in DS patients is significantly higher than the general adult population of São Paulo, Brazil, in a similar age, between 20–29 years and 30–39 years [51]. The prevalence in these healthy adult individuals was 7.4% and 24.2%, respectively [51]. Also, in the near future, this high prevalence of OSA can reduce in DS patients with early stimulation that includes oropharyngeal exercises in DS [52,53], which have recently been described as essential measures to decrease OSA-related symptoms in the population general [54–58]. These exercises improve OSA by reducing the circumference of mouth and neck breathing and improving lip hypotonia, the resting position of the tongue, and hypotonia [54–58]. All these aspects are critical points in the population with DS. The impact of these exercises should be more important in childhood. In general, there is an improvement in the severity of OSA throughout childhood, and the earlier the diagnosis and treatment of OSA is started, the less the possible damage to health, and the better the general quality of life and longevity of people with DS [10]. New studies in the future should address the role of oropharyngeal exercises in DS in childhood and adult.

In addition, since we do not have DS patients without OSA, we cannot extend the Stop-Bang score for these patients. Nevertheless, SBQ can be a useful tool to exclude patients without moderate-severe OSA in the DS population, since negative SBQ (score < 3) was not

observed in moderate-severe OSA. In contrast, we observed positive SBQ (score $\geq$ 3) in 6 adults with mild OSA. Consequently, it is plausible that positive SBQ may occur in DS patients without OSA. However, as this is the first study evaluating the role of the SBQ for the screening of moderate-severe OSA in DS adults, further studies are needed for external validation of our findings. Notwithstanding, the main finding of the study is that the application of the SBQ can select the individuals at higher risk for moderate-severe OSA who should undergo polysomnography. This result is significant since access to polysomnography is limited in the majority of the countries.

Considering that the treatment of moderate-severe OSA indicates specific interventions such as the use of positive airway pressure (PAP) or even some type of surgical intervention, the identification of these patients is essential to ensure an improvement in the quality of life, longevity, and reduction of morbidities associated with OSA in people with DS. [10]

## Conclusion

In conclusion, our study confirmed the very high prevalence of OSA in adults with DS. Additionally, the SBQ, hematocrit levels and BMI showed a strong correlation with OSA severity. Furthermore, the SBQ showed high sensitivity and specificity in identifying moderate-severe OSA in this population. Taken together, these results open new horizons to screening OSA in adults with DS. Nevertheless, considering the limited access to polysomnography for all patients, the use of hematocrit levels and BMI, and especially the application of the SBQ can select the individuals at higher risk for moderate-severe OSA who should undergo the polysomnography. However, further studies with a larger population of patients with DS are needed for more definitive conclusions to be drawn.

## Supporting information

**S1 Data.**
(XLS)

**S2 Data.**
(XLS)

## Acknowledgments

The authors would like to thank the study participants and their proxies for their patience and contributions to the research.

## Author Contributions

**Conceptualization:** Anderson Albuquerque de Carvalho, Fabio Ferreira Amorim, Levy Aniceto Santana, Karlo Jozefo Quadros de Almeida, Alfredo Nicodemos Cruz Santana, Francisco de Assis Rocha Neves.

**Data curation:** Anderson Albuquerque de Carvalho.

**Formal analysis:** Anderson Albuquerque de Carvalho, Fabio Ferreira Amorim, Alfredo Nicodemos Cruz Santana.

**Funding acquisition:** Levy Aniceto Santana.

**Investigation:** Anderson Albuquerque de Carvalho.

**Methodology:** Anderson Albuquerque de Carvalho, Karlo Jozefo Quadros de Almeida, Alfredo Nicodemos Cruz Santana, Francisco de Assis Rocha Neves.

**Project administration:** Anderson Albuquerque de Carvalho, Alfredo Nicodemos Cruz Santana, Francisco de Assis Rocha Neves.

**Resources:** Anderson Albuquerque de Carvalho, Fabio Ferreira Amorim, Levy Aniceto Santana, Karlo Jozefo Quadros de Almeida, Alfredo Nicodemos Cruz Santana, Francisco de Assis Rocha Neves.

**Supervision:** Alfredo Nicodemos Cruz Santana, Francisco de Assis Rocha Neves.

**Validation:** Anderson Albuquerque de Carvalho, Fabio Ferreira Amorim, Alfredo Nicodemos Cruz Santana, Francisco de Assis Rocha Neves.

**Visualization:** Anderson Albuquerque de Carvalho, Fabio Ferreira Amorim, Levy Aniceto Santana, Karlo Jozefo Quadros de Almeida, Alfredo Nicodemos Cruz Santana, Francisco de Assis Rocha Neves.

**Writing – original draft:** Anderson Albuquerque de Carvalho.

**Writing – review & editing:** Fabio Ferreira Amorim, Levy Aniceto Santana, Karlo Jozefo Quadros de Almeida, Alfredo Nicodemos Cruz Santana, Francisco de Assis Rocha Neves.

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
