## [Decision Letter · Decision Letter 0]

27 Jan 2020

PONE-D-19-26496

STOP-Bang questionnaire should be used in all adults with Down Syndrome to screen for moderate to severe obstructive sleep apnea

PLOS ONE

Dear Mr Carvalho,

Thank you for submitting your manuscript to PLOS ONE. After careful consideration, we feel that it has merit but does not fully meet PLOS ONE’s publication criteria as it currently stands. Therefore, we invite you to submit a revised version of the manuscript that addresses the points raised during the review process.

ACADEMIC EDITOR: The main limitation is the lack of patient without OSA. The authors should explain this issue and attempt to correlate OSAS severity and Stop Bang score.

We would appreciate receiving your revised manuscript by Mar 12 2020 11:59PM. To enhance the reproducibility of your results, we recommend that if applicable you deposit your laboratory protocols in protocols.io, where a protocol can be assigned its own identifier (DOI) such that it can be cited independently in the future. For instructions see: http://journals.plos.org/plosone/s/submission-guidelines#loc-laboratory-protocols

We look forward to receiving your revised manuscript.

Kind regards,

Andrea Romigi, M.D., Ph.D

Academic Editor

PLOS ONE

Journal Requirements:

Reviewers' comments:

Reviewer's Responses to Questions

**Comments to the Author**

1. Is the manuscript technically sound, and do the data support the conclusions?

Reviewer #1: Yes

Reviewer #2: Yes

2. Has the statistical analysis been performed appropriately and rigorously? 

Reviewer #1: Yes

Reviewer #2: Yes

3. Have the authors made all data underlying the findings in their manuscript fully available?

Reviewer #1: Yes

Reviewer #2: Yes

4. Is the manuscript presented in an intelligible fashion and written in standard English?

Reviewer #1: Yes

Reviewer #2: Yes

5. Review Comments to the Author

Reviewer #1: This well written article reports the results of a study aimed at assessing the real incidence of OSA in adults with DS and at investigating whether biochemical analysis and/or questionnaires could help in identifying subjects at higher risk for OSA.

Overall, this paper is well written and well organized, and could certainly be of interest for the readers of the Journal.

I only have a couple of minor points.

There is a brief mention to screening for OSA during childhood. Have the Authors some data on the “stability” of this diagnosis? Are there children with DS and OSA presenting a significant improvement of their symptoms? And how many adults already presented with symptoms during childhood? And in these case, were they recognized, or misdiagnosed? Some of this information could be added for the present series.

A second point is about the percentage of OSA found in this study. Reporting a 100% prevalence of OSA limits the strength of the SBQ as a possible screening test, since it would be interesting to understand the difference in scores between adults with and without OSA, and not only reporting a different severity. This limit should be pointed out and discussed.

Reviewer #2: this is a very nice manuscript examining the utility of the stop-bang questionnaire in individuals with trisomy 21. overall, it is well conceived and carried out, and adds to the literature.

a few points:

1. a major limitation is the lack of individuals without osa in the sample.

2. given #1, i would suggest to the authors that they emphasize the role of the stop-bang in potentially differentiating mild from mod/severe osa in their sample, as there may be differences in associated comorbidity.

3. in the results section, especially the last paragraph, recommend being very clear about what the predictive values are actually predicting (i.e. mod/severe osa or just any osa?).

6. PLOS authors have the option to publish the peer review history of their article (what does this mean?). If published, this will include your full peer review and any attached files.

Reviewer #1: No

Reviewer #2: No

---

## [Author Response · Author response to Decision Letter 0]

3 Mar 2020

To Andrea Romigi, MD, Ph.D 

Academic Editor 

PLOS ONE

Dear Editor

Thank you for considering the paper: “STOP-Bang questionnaire should be used in all adults with Down Syndrome to screen for moderate to severe obstructive sleep apnea” (PONE-D-19-26496)” by Carvalho et al., for publication in the PLOS ONE.

Thank you for allowing us to re-submit this revised manuscript.

We would also like to thank all the reviewers for their positive comments and helpful criticisms of our paper. 

We hope that we answered all the queries, which certainly improved the quality of the manuscript, making the article more suitable to contribute to the relevance of OSA in adults with Down Syndrome (DS). 

We are looking forward to your decision regarding the suitability of the revised version of this paper for the Journal.

Thank you very much in advance,

Sincerely,

Anderson Albuquerque de Carvalho

AUTHORS’ REPLY TO REVIEWER’S #1 COMMENTS

Q- 1. There is a brief mention to screening for OSA during childhood. Have the Authors some data on the “stability” of this diagnosis? Are there children with DS and OSA presenting a significant improvement of their symptoms? 

Answer to query Q1:

We thank the Reviewer for constructive criticism. No, unfortunately, we do not have any data for OSA in childhood. On the other hand, we think that is an important issue to be discussed in the manuscript. 

Concerning the "stability" of the diagnosis in children with DS, in a recent meta-analysis, the prevalence and severity of OSA in children with DS that OSA (AHI> 1, 1.5 or 2 events / h) was highly prevalent, affecting 69% to 76% of children with DS[9]. Besides, approximately half of these children (50%) had moderate to severe OSA (AHI> 5 events / h). In particular, this meta-analysis concluded that some children with DS had moderate to severe OSA at their young ages.

Concerning whether children with DS and OSA present a significant improvement in their symptoms, some studies addressed this question. In general, there is an improvement in the severity of OSA throughout childhood and, the earlier the diagnosis and treatment of OSA is started, the less the possible damage to health, the better the general quality of life and longevity of people with DS[10]. Soon, this high prevalence of OSA in DS patients, even in childhood, may be reduced with early stimulation that includes oropharyngeal exercises in DS [35,36], which has recently been used as essential measures for problems related to OSA in the general population[37-41]. These exercises improve OSA by reducing the circumference of mouth breathing and the neck and improving lip hypotonia, the resting position of the tongue, and hypotonia[37-41]. All these aspects are critical points in the population with DS.

We have included a paragraph in the text that provides information on the prevalence and improvement of symptoms of OSA in children with DS. The new text is marked in yellow and was inserted in the introduction and discussion.

Introduction

“A common comorbidity in people with DS is obstructive sleep apnea (OSA), whose prevalence ranges from 78 to 100% in adults[3-8] and 69% to 76% in children [9].” 

Discussion

“Also, in the near future, this high prevalence of OSA can reduce in DS patients with early stimulation that includes oropharyngeal exercises in DS[35,36], which have recently been described as important measures to decrease OSA-related symptoms in the population general[37-41]. These exercises improve OSA by reducing the circumference of mouth and neck breathing and improving lip hypotonia, the resting position of the tongue, and hypotonia[37-41]. All these aspects are critical points in the population with DS. The impact of these exercises should be more important in childhood. In general, there is an improvement in the severity of OSA throughout childhood, and the earlier the diagnosis and treatment of OSA is started, the less the possible damage to health, and the better the general quality of life and longevity of people with DS[10]. New studies in the future should address the role of oropharyngeal exercises in DS in childhood and adult.”

Q- 2. And how many adults already presented with symptoms during childhood? And in these case, were they recognized, or misdiagnosed? Some of this information could be added for the present series. 

Answer to query Q2:

We thank the reviewer for this observation. This is another very important issue to be studied. However, we did not investigate the symptoms and diagnosis of OSA in childhood. In fact, we did not find in the literature studies carried out in the adult population with DS that investigated the presence of OSA symptoms in childhood. Notwithstanding, that is a good idea and, since we continue the study enrolling new patients, we will add this question in the interview. Unfortunately, we did not have a sleep disorder unit for childhood. We are trying to set up this unit and, your comment encourages us to continue our demand to the Brazilian authorities. 

Q- 3. A second point is about the percentage of OSA found in this study. Reporting a 100% prevalence of OSA limits the strength of the SBQ as a possible screening test, since it would be interesting to understand the difference in scores between adults with and without OSA, and not only reporting a different severity. This limit should be pointed out and discussed.

Answer to query Q3:

We thank the reviewer for the constructive comments. We agree that the absence of adults with DS without OSA is the main limitation of this study. Since the present study was performed in a single center for DS patients from a reference hospital, it is possible that our results may represent a specific population of DS adults, with more comorbidities, since DS patients treated in a community care institution were not included. Moreover, considering that the prevalence of DS patients without OSA should be quite low, It will be necessary to increase the sample size, and, at some point, a small number of adults without OSA would be identified. Interestingly, so far, the largest study that evaluated OSA in adults DS patients included 47 individuals[5] reported a prevalence of 78% of OSA. Therefore, the high prevalence of OSA observed in our DS patients is similar to what was described in other studies, which varies between 78-100%[3-8]. It is conceivable that the absence of adults with DS without OSA is possibly related to the high prevalence of overweight and obesity observed in our patients, which was 68%. Nevertheless, a similar prevalence of overweight and obesity was described by other studies[5,57].

The prevalence of OSA observed in the present study in DS patients is significantly higher than the general adult population of São Paulo, Brazil, in a similar age, between 20-29 years and 30-39 years [58]. The prevalence in these typical adults individual was 7.4% and 24.2%, respectively[58]. Also, in the near future, this high prevalence of OSA can reduce in DS patients with early stimulation that includes oropharyngeal exercises in DS[35,36], which have recently been described as important measures to decrease OSA-related symptoms in the population general[37-41]. These exercises improve OSA by reducing the circumference of mouth and neck breathing and improving lip hypotonia, the resting position of the tongue, and hypotonia[37-41]. Concerning the comment: “Reporting a 100% prevalence of OSA limits the strength of the SBQ as a possible screening test since it would be interesting to understand the difference in scores between adults with and without OSA, and not only reporting a different severity.” We agree that this is another limitation of our study. Since we do not have DS patients without OSA, we cannot extend the Stop-Bang score for these patients. Nevertheless, SBQ can be a useful tool to exclude patients without moderate-severe OSA in the DS population, since negative SBQ (score < 3) was not observed in moderate-severe OSA. In contrast, we observed positive SBQ (score > 3) in 6 patients with mild OSA. Consequently, it is plausible that positive SBQ may occur in DS patients without OSA. 

However, the main finding of the study is that the application of the SBQ can select the individuals at higher risk for moderate-severe OSA who should undergo polysomnography. This result is very important since access to polysomnography is limited in the majority of the countries. 

Considering that the treatment of moderate-severe OSA indicates specific interventions such as the use of positive airway pressure (PAP) or even some type of surgical intervention, the identification of these patients is essential to ensure an improvement in the quality of life, longevity, and reduction of morbidities associated with OSA in people with DS.[10]

To improve the quality of the manuscript, we included all these observations in the discussion/conclusion section when we wrote the limitations of the study. They are marked in yellow. 

Discussion 

Our study shows some limitations. First, the individuals were submitted to a type III PSG and not to a full PSG test. Consequently, we did not have information related to sleep architecture, and OSA severity (REI) may have been underestimated. However, the type III PSG has been used in DS with good results.[23,50-52] Moreover, the number of technical problems that required the type III PSG to be repeated was small (10% in our study), which is in accordance to previous studies in DS or not (3–18%).[22,23,54-56]

“Another limitation of the present study is the absence of DS adults without OSA. Since the present study was performed in a single center for DS patients from a reference hospital and did not include patients treated in a community care institution, our results may represent a specific population of DS adults, with more comorbidities. Moreover, considering that the prevalence of DS patients without OSA should be quite low, it will be necessary to increase the sample size, and, at some point, a small number of adults without OSA would be identified. Interestingly, so far, the largest study that evaluated OSA in adults DS patients included 47 individuals[5] reported a prevalence of 78% of OSA. Therefore, the high prevalence of OSA observed in our DS patients is similar to what was described in other studies, which varies between 78-100%[3-8]. It is conceivable that the absence of adults with DS without OSA is possibly related to the high prevalence of overweight and obesity observed in our patients, which was 68%. Nevertheless, a similar prevalence of overweight and obesity was described by other studies[5,57]. The prevalence of OSA observed in the present study in DS patients is significantly higher than the general adult population of São Paulo, Brazil, in a similar age, between 20-29 years and 30-39 years [58]. The prevalence in these healthy adult individuals was 7.4% and 24.2%, respectively[58]. Also, in the near future, this high prevalence of OSA can reduce in DS patients with early stimulation that includes oropharyngeal exercises in DS[35,36], which have recently been described as essential measures to decrease OSA-related symptoms in the population general[37-41]. These exercises improve OSA by reducing the circumference of mouth and neck breathing and improving lip hypotonia, the resting position of the tongue, and hypotonia[37-41]. All these aspects are critical points in the population with DS. The impact of these exercises should be more important in childhood. In general, there is an improvement in the severity of OSA throughout childhood, and the earlier the diagnosis and treatment of OSA is started, the less the possible damage to health, and the better the general quality of life and longevity of people with DS[10]. New studies in the future should address the role of oropharyngeal exercises in DS in childhood and adult.” 

“In addition, since we do not have DS patients without OSA, we cannot extend the Stop-Bang score for these patients. Nevertheless, SBQ can be a useful tool to exclude patients without moderate-severe OSA in the DS population, since negative SBQ (score < 3) was not observed in moderate-severe OSA. In contrast, we observed positive SBQ (score ≥ 3) in 6 adults with mild OSA. Consequently, it is plausible that positive SBQ may occur in DS patients without OSA. However, as this is the first study evaluating the role of the SBQ for the screening of moderate-severe OSA in DS adults, further studies are needed for external validation of our findings. Notwithstanding, the main finding of the study is that the application of the SBQ can select the individuals at higher risk for moderate-severe OSA who should undergo polysomnography. This result is significant since access to polysomnography is limited in the majority of the countries.”

“Considering that the treatment of moderate-severe OSA indicates specific interventions such as the use of positive airway pressure (PAP) or even some type of surgical intervention, the identification of these patients is essential to ensure an improvement in the quality of life, longevity, and reduction of morbidities associated with OSA in people with DS.[10]”

Conclusion

“Furthermore, the SBQ showed high sensitivity and specificity in identifying moderate-severe OSA in this population. Taken together, these results open new horizons to screening OSA in adults with DS.”

AUTHORS’ REPLY TO REVIEWER’S #2 COMMENTS

Q- 1. a major limitation is the lack of individuals without osa in the sample. 

Answer to query Q1:

We thank the reviewer for the constructive comments. We agree that the absence of adults with DS without OSA is the main limitation of this study. Since the present study was performed in a single center for DS patients from a reference hospital, it is possible that our results may represent a specific population of DS adults, with more comorbidities, since DS patients treated in a community care institution were not included. Moreover, considering that the prevalence of DS patients without OSA should be quite low, It will be necessary to increase the sample size, and, at some point, a small number of adults without OSA would be identified. Interestingly, so far, the largest study that evaluated OSA in adults DS patients included 47 individuals[5] reported a prevalence of 78% of OSA. Therefore, the high prevalence of OSA observed in our DS patients is similar to what was described in other studies, which varies between 78-100%[3-8]. It is conceivable that the absence of adults with DS without OSA is possibly related to the high prevalence of overweight and obesity observed in our patients, which was 68%. Nevertheless, a similar prevalence of overweight and obesity was described by other studies[5,57].

The prevalence of OSA observed in the present study in DS patients is significantly higher than the general adult population of São Paulo, Brazil, in a similar age, between 20-29 years and 30-39 years [58]. The prevalence in these typical adults individual was 7.4% and 24.2%, respectively[58]. Also, in the near future, this high prevalence of OSA can reduce in DS patients with early stimulation that includes oropharyngeal exercises in DS[35,36], which have recently been described as important measures to decrease OSA-related symptoms in the population general[37-41]. These exercises improve OSA by reducing the circumference of mouth and neck breathing and improving lip hypotonia, the resting position of the tongue, and hypotonia[37-41]. Concerning the comment: “Reporting a 100% prevalence of OSA limits the strength of the SBQ as a possible screening test since it would be interesting to understand the difference in scores between adults with and without OSA, and not only reporting a different severity.” We agree that this is another limitation of our study. Since we do not have DS patients without OSA, we cannot extend the Stop-Bang score for these patients. Nevertheless, SBQ can be a useful tool to exclude patients without moderate-severe OSA in the DS population, since negative SBQ (score < 3) was not observed in moderate-severe OSA. In contrast, we observed positive SBQ (score > 3) in 6 patients with mild OSA. Consequently, it is plausible that positive SBQ may occur in DS patients without OSA. 

However, the main finding of the study is that the application of the SBQ can select the individuals at higher risk for moderate-severe OSA who should undergo polysomnography. This result is very important since access to polysomnography is limited in the majority of the countries. 

Considering that the treatment of moderate-severe OSA indicates specific interventions such as the use of positive airway pressure (PAP) or even some type of surgical intervention, the identification of these patients is essential to ensure an improvement in the quality of life, longevity, and reduction of morbidities associated with OSA in people with DS.[10]

To improve the quality of the manuscript, we included all these observations in the discussion/conclusion section when we wrote the limitations of the study. They are marked in yellow. 

Discussion 

Our study shows some limitations. First, the individuals were submitted to a type III PSG and not to a full PSG test. Consequently, we did not have information related to sleep architecture, and OSA severity (REI) may have been underestimated. However, the type III PSG has been used in DS with good results.[23,50-52] Moreover, the number of technical problems that required the type III PSG to be repeated was small (10% in our study), which is in accordance to previous studies in DS or not (3–18%).[22,23,54-56]

“Another limitation of the present study is the absence of DS adults without OSA. Since the present study was performed in a single center for DS patients from a reference hospital and did not include patients treated in a community care institution, our results may represent a specific population of DS adults, with more comorbidities. Moreover, considering that the prevalence of DS patients without OSA should be quite low, it will be necessary to increase the sample size, and, at some point, a small number of adults without OSA would be identified. Interestingly, so far, the largest study that evaluated OSA in adults DS patients included 47 individuals[5] reported a prevalence of 78% of OSA. Therefore, the high prevalence of OSA observed in our DS patients is similar to what was described in other studies, which varies between 78-100%[3-8]. It is conceivable that the absence of adults with DS without OSA is possibly related to the high prevalence of overweight and obesity observed in our patients, which was 68%. Nevertheless, a similar prevalence of overweight and obesity was described by other studies[5,57]. The prevalence of OSA observed in the present study in DS patients is significantly higher than the general adult population of São Paulo, Brazil, in a similar age, between 20-29 years and 30-39 years [58]. The prevalence in these healthy adult individuals was 7.4% and 24.2%, respectively[58]. Also, in the near future, this high prevalence of OSA can reduce in DS patients with early stimulation that includes oropharyngeal exercises in DS[35,36], which have recently been described as essential measures to decrease OSA-related symptoms in the population general[37-41]. These exercises improve OSA by reducing the circumference of mouth and neck breathing and improving lip hypotonia, the resting position of the tongue, and hypotonia[37-41]. All these aspects are critical points in the population with DS. The impact of these exercises should be more important in childhood. In general, there is an improvement in the severity of OSA throughout childhood, and the earlier the diagnosis and treatment of OSA is started, the less the possible damage to health, and the better the general quality of life and longevity of people with DS[10]. New studies in the future should address the role of oropharyngeal exercises in DS in childhood and adult.” 

“In addition, since we do not have DS patients without OSA, we cannot extend the Stop-Bang score for these patients. Nevertheless, SBQ can be a useful tool to exclude patients without moderate-severe OSA in the DS population, since negative SBQ (score < 3) was not observed in moderate-severe OSA. In contrast, we observed positive SBQ (score ≥ 3) in 6 adults with mild OSA. Consequently, it is plausible that positive SBQ may occur in DS patients without OSA. However, as this is the first study evaluating the role of the SBQ for the screening of moderate-severe OSA in DS adults, further studies are needed for external validation of our findings. Notwithstanding, the main finding of the study is that the application of the SBQ can select the individuals at higher risk for moderate-severe OSA who should undergo polysomnography. This result is significant since access to polysomnography is limited in the majority of the countries.”

“Considering that the treatment of moderate-severe OSA indicates specific interventions such as the use of positive airway pressure (PAP) or even some type of surgical intervention, the identification of these patients is essential to ensure an improvement in the quality of life, longevity, and reduction of morbidities associated with OSA in people with DS.[10]”

Conclusion

“Furthermore, the SBQ showed high sensitivity and specificity in identifying moderate-severe OSA in this population. Taken together, these results open new horizons to screening OSA in adults with DS.”

Q- 2. given #1, i would suggest to the authors that they emphasize the role of the stop-bang in potentially differentiating mild from mod/severe osa in their sample, as there may be differences in associated comorbidity. 

Answer to query Q2: 

We thank the Reviewer for this observation. We agree and changed the text in the manuscript that emphasize this observation and makes the text clearer to the reader. These modifications are in yellow.

Discussion

“The present study contributes to important findings. First, when compared to previously published studies[3-8], the present included the largest number of adults with DS. Second, it evaluated for the first time the role of the SBQ in moderate-severe OSA screening in this population.”

“Besides, our study also showed the relation between higher SBQ score and higher REI (OSA severity) in DS adults. This relation has been widely shown in the general population[45-47]. As an example, a recent meta-analysis study demonstrated that the higher the SBQ score, the greater the likelihood of moderate-severe OSA[45].”

“Regarding the use of sleep questionnaires for OSA screening, our study was the first to identify a questionnaire capable of playing a significant role in moderate-severe OSA screening in DS adults.”

Q- 3. in the results section, especially the last paragraph, recommend being very clear about what the predictive values are actually predicting (i.e. mod/severe osa or just any osa?)

Answer to query Q3:

We thank the reviewer for this observation. You are right, and we made changes in the manuscript to turn this aspect more precise. 

In Results, we emphasize in the text of the study when the information is related to moderate-severe OSA. 

Results

“As the SBQ remained in the final multiple linear regression model, the contingency table analysis (2x2) with moderate-severe OSA was performed. Positive SBQ (3 or more affirmative answers) showed in the screening of moderate-severe OSA, 100% of sensitivity (95%CI: 92.75-100%), 45.45% of specificity (95%CI: 16.75-76.62%), 89.09% of PPV (95%CI: 82.64-93.34%), 100% of NPV, 90% of accuracy (95%CI: 79.49-96.24%) and OR of 24.29.”

AUTHORS’ REPLY TO ACADEMIC EDITOR 

Q- 1. The main limitation is the lack of patient without OSA. The authors should explain this issue and attempt to correlate OSAS severity and Stop Bang score. 

Answer to query Q1:

We thank the Academic Editor for the constructive comments. We agree that the absence of adults with DS without OSA is the main limitation of this study. Since the present study was performed in a single center for DS patients from a reference hospital, it is possible that our results may represent a specific population of DS adults, with more comorbidities, since DS patients treated in a community care institution were not included. Moreover, considering that the prevalence of DS patients without OSA should be quite low, It will be necessary to increase the sample size, and, at some point, a small number of adults without OSA would be identified. Interestingly, so far, the largest study that evaluated OSA in adults DS patients included 47 individuals[5] reported a prevalence of 78% of OSA. Therefore, the high prevalence of OSA observed in our DS patients is similar to what was described in other studies, which varies between 78-100%[3-8]. It is conceivable that the absence of adults with DS without OSA is possibly related to the high prevalence of overweight and obesity observed in our patients, which was 68%. Nevertheless, a similar prevalence of overweight and obesity was described by other studies[5,57].

The prevalence of OSA observed in the present study in DS patients is significantly higher than the general adult population of São Paulo, Brazil, in a similar age, between 20-29 years and 30-39 years [58]. The prevalence in these typical adults individual was 7.4% and 24.2%, respectively[58]. Also, in the near future, this high prevalence of OSA can reduce in DS patients with early stimulation that includes oropharyngeal exercises in DS[35,36], which have recently been described as important measures to decrease OSA-related symptoms in the population general[37-41]. These exercises improve OSA by reducing the circumference of mouth and neck breathing and improving lip hypotonia, the resting position of the tongue, and hypotonia[37-41]. Concerning the comment: “Reporting a 100% prevalence of OSA limits the strength of the SBQ as a possible screening test since it would be interesting to understand the difference in scores between adults with and without OSA, and not only reporting a different severity.” We agree that this is another limitation of our study. Since we do not have DS patients without OSA, we cannot extend the Stop-Bang score for these patients. Nevertheless, SBQ can be a useful tool to exclude patients without moderate-severe OSA in the DS population, since negative SBQ (score < 3) was not observed in moderate-severe OSA. In contrast, we observed positive SBQ (score > 3) in 6 patients with mild OSA. Consequently, it is plausible that positive SBQ may occur in DS patients without OSA. 

However, the main finding of the study is that the application of the SBQ can select the individuals at higher risk for moderate-severe OSA who should undergo polysomnography. This result is very important since access to polysomnography is limited in the majority of the countries. 

Considering that the treatment of moderate-severe OSA indicates specific interventions such as the use of positive airway pressure (PAP) or even some type of surgical intervention, the identification of these patients is essential to ensure an improvement in the quality of life, longevity, and reduction of morbidities associated with OSA in people with DS.[10]

To improve the quality of the manuscript, we included all these observations in the discussion/conclusion section when we wrote the limitations of the study. They are marked in yellow. 

Discussion 

Our study shows some limitations. First, the individuals were submitted to a type III PSG and not to a full PSG test. Consequently, we did not have information related to sleep architecture, and OSA severity (REI) may have been underestimated. However, the type III PSG has been used in DS with good results.[23,50-52] Moreover, the number of technical problems that required the type III PSG to be repeated was small (10% in our study), which is in accordance to previous studies in DS or not (3–18%).[22,23,54-56]

“Another limitation of the present study is the absence of DS adults without OSA. Since the present study was performed in a single center for DS patients from a reference hospital and did not include patients treated in a community care institution, our results may represent a specific population of DS adults, with more comorbidities. Moreover, considering that the prevalence of DS patients without OSA should be quite low, it will be necessary to increase the sample size, and, at some point, a small number of adults without OSA would be identified. Interestingly, so far, the largest study that evaluated OSA in adults DS patients included 47 individuals[5] reported a prevalence of 78% of OSA. Therefore, the high prevalence of OSA observed in our DS patients is similar to what was described in other studies, which varies between 78-100%[3-8]. It is conceivable that the absence of adults with DS without OSA is possibly related to the high prevalence of overweight and obesity observed in our patients, which was 68%. Nevertheless, a similar prevalence of overweight and obesity was described by other studies[5,57]. The prevalence of OSA observed in the present study in DS patients is significantly higher than the general adult population of São Paulo, Brazil, in a similar age, between 20-29 years and 30-39 years [58]. The prevalence in these healthy adult individuals was 7.4% and 24.2%, respectively[58]. Also, in the near future, this high prevalence of OSA can reduce in DS patients with early stimulation that includes oropharyngeal exercises in DS[35,36], which have recently been described as essential measures to decrease OSA-related symptoms in the population general[37-41]. These exercises improve OSA by reducing the circumference of mouth and neck breathing and improving lip hypotonia, the resting position of the tongue, and hypotonia[37-41]. All these aspects are critical points in the population with DS. The impact of these exercises should be more important in childhood. In general, there is an improvement in the severity of OSA throughout childhood, and the earlier the diagnosis and treatment of OSA is started, the less the possible damage to health, and the better the general quality of life and longevity of people with DS[10]. New studies in the future should address the role of oropharyngeal exercises in DS in childhood and adult.” 

“In addition, since we do not have DS patients without OSA, we cannot extend the Stop-Bang score for these patients. Nevertheless, SBQ can be a useful tool to exclude patients without moderate-severe OSA in the DS population, since negative SBQ (score < 3) was not observed in moderate-severe OSA. In contrast, we observed positive SBQ (score ≥ 3) in 6 adults with mild OSA. Consequently, it is plausible that positive SBQ may occur in DS patients without OSA. However, as this is the first study evaluating the role of the SBQ for the screening of moderate-severe OSA in DS adults, further studies are needed for external validation of our findings. Notwithstanding, the main finding of the study is that the application of the SBQ can select the individuals at higher risk for moderate-severe OSA who should undergo polysomnography. This result is significant since access to polysomnography is limited in the majority of the countries.”

“Considering that the treatment of moderate-severe OSA indicates specific interventions such as the use of positive airway pressure (PAP) or even some type of surgical intervention, the identification of these patients is essential to ensure an improvement in the quality of life, longevity, and reduction of morbidities associated with OSA in people with DS.[10]”

Conclusion

“Furthermore, the SBQ showed high sensitivity and specificity in identifying moderate-severe OSA in this population. Taken together, these results open new horizons to screening OSA in adults with DS.”

---

## [Decision Letter · Decision Letter 1]

20 Apr 2020

STOP-Bang questionnaire should be used in all adults with Down Syndrome to screen for moderate to severe obstructive sleep apnea

PONE-D-19-26496R1

Dear Dr. Carvalho,

We are pleased to inform you that your manuscript has been judged scientifically suitable for publication and will be formally accepted for publication once it complies with all outstanding technical requirements.

With kind regards,

Andrea Romigi, M.D., Ph.D

Academic Editor

PLOS ONE

Additional Editor Comments (optional):

Reviewers' comments:

Reviewer's Responses to Questions

**Comments to the Author**

1. If the authors have adequately addressed your comments raised in a previous round of review and you feel that this manuscript is now acceptable for publication, you may indicate that here to bypass the “Comments to the Author” section, enter your conflict of interest statement in the “Confidential to Editor” section, and submit your "Accept" recommendation.

Reviewer #1: All comments have been addressed

2. Is the manuscript technically sound, and do the data support the conclusions?

Reviewer #1: Yes

3. Has the statistical analysis been performed appropriately and rigorously? 

Reviewer #1: Yes

4. Have the authors made all data underlying the findings in their manuscript fully available?

Reviewer #1: Yes

5. Is the manuscript presented in an intelligible fashion and written in standard English?

Reviewer #1: Yes

6. Review Comments to the Author

Reviewer #1: The Authors answered to the raised queries, The article can now be accepted as it is without further modification

7. PLOS authors have the option to publish the peer review history of their article (what does this mean?). If published, this will include your full peer review and any attached files.

Reviewer #1: No

---

## [Editor Report · Acceptance letter]

22 Apr 2020

PONE-D-19-26496R1 

STOP-Bang questionnaire should be used in all adults with Down Syndrome to screen for moderate to severe obstructive sleep apnea 

Dear Dr. Carvalho:

I am pleased to inform you that your manuscript has been deemed suitable for publication in PLOS ONE. Congratulations! Your manuscript is now with our production department. 

With kind regards,

on behalf of

Dr. Andrea Romigi 

Academic Editor

PLOS ONE